

# Fish nursery value of algae habitats in temperate coastal reefs

Hilmar Hinz[1,2], Olga Reñones[2], Adam Gouraguine[3], Andrew F. Johnson[4] and Joan Moranta[2]

[1] Instituto Mediterraneo de Estudios Avanzados (IMEDEA; CSIC-UIB), Esporles, Illes Balears, Spain
[2] Instituto Español de Oceanografía (IEO), Centre Oceanogràfic de les Balears, Ecosystem Oceanography Group (GRECO), Palma, Illes Balears, Spain
[3] School of Biological Sciences, University of Essex, Colchester, United Kingdom
[4] MarFishEco, Portland, OR, United States of America

Corresponding author
Hilmar Hinz,
hhinz@imedea.uib-csic.es

## ABSTRACT

The nursery function of coastal habitats is one of the most frequently mentioned and recognized ecosystem services in the valuation of coastal ecosystems. Despite its importance our understanding of the precise habitat parameters and mechanisms that make a habitat important as a nursery area is still limited for many species. The study aimed to establish the importance of different algae morphotypes in providing shelter and food for juvenile coastal fish during the main settlement peaks, in early spring and late summer, in littoral rocky reef systems in the Northwestern Mediterranean. The results of our study showed strong seasonal differences in algae cover, composition and height between the two sampling periods. Overall, during spring the algae were well developed, while in late summer, both density and height, of most algae decreased considerably. Equally, prey biomass, in form of suitable sized invertebrate fauna associated to the algae, decreased. Accordingly, the shelter and food for the fish settling in this habitat during late summer were less abundant, indicating a mismatch between the observed presence of juvenile fish and optimal habitat conditions. Differences in prey densities were detected between algae morphotypes, with structurally more complex algae, such as *Cystoseira* spp. and *Halopteris* spp. consistently containing more prey, independent of season, compared to simpler structured morphotypes such as Dictoytales. The study furthermore related juvenile fish density to habitats dominated by different algae morphotypes. Out of the three-study species (*Diplodus vulgaris, Symphodus ocellatus, Coris julis*) only *S. ocellatus* showed a significant association with an algae habitat. *S. ocellatus* related positively to habitats dominated by Dictoytales which provided the highest cover during late summer but had the lowest prey densities. A strong association of this species with *Cystoseira*, as reported by other studies, could not be confirmed. *Cystoseira* was abundant within the study area but in a state of dieback, showing loss and reduced height of foliage, typical for the time of year within the study area. It is therefore likely that algae-fish associations are context-dependent and that several algae species may fulfil similar functions. We also discovered that prey biomass did not appear to have an important effect on juvenile abundances. Nevertheless, the availability of prey may influence juvenile fish condition, growth performance and ultimately long-term survival. We therefore suggest that future studies on habitat quality should also include, besides abundance, indicators related to the condition and growth of juveniles.

## INTRODUCTION

The provision and value of nursery habitats by the coastal zone is one of the most frequently mentioned and recognized ecosystem services in the valuation of coastal marine ecosystems (*Duarte, 2000*; *Jackson et al., 2001*; *Jackson et al., 2015*). As many coastal habitats are under increasing human pressures from urbanisation, fishing, climatic change and the introduction of alien species (*Sala et al., 2011*; *Sala et al., 2012*; *Vergés et al., 2014a*; *Vergés et al., 2014b*), the provisioning of this ecosystem service is progressively also under threat. While the large-scale importance of nursery habitats for ecosystem functioning, food production and integrity of ecosystems is recognized, a detailed understanding over which habitat types constitute to this function is still lacking (*Beck et al., 2001*). Seagrass meadows and estuarine systems have been the focus of marine nursery habitat research (e.g., *Heck, Hays & Orth, 2003*; *Seitz et al., 2013*; *Woodland et al., 2012*; *Ruiz-Frau et al., 2017*), while studies that focus on littoral rocky reef systems appear less frequently in the literature (but see *Cheminée et al., 2017*; *Guidetti, 2000*; *Harmelin-Vivien, Harmelin & Leboulleux, 1995*). Although spatially less extensive compared to seagrass meadows, littoral rocky reef systems harbour a wide variety of different algae species, providing shelter and food, in the form of associated fauna, to juvenile fish (*Harmelin-Vivien, Harmelin & Leboulleux, 1995*; *Cheminée et al., 2013*; *Félix-Hackradt et al., 2014*).

In the Mediterranean, littoral rocky reef habitats are used by a variety of commercial and non-commercial species for part, or their entire life cycle (*Harmelin-Vivien, Harmelin & Leboulleux, 1995*; *Guidetti, 2000*; *La Mesa et al., 2011*; *Félix-Hackradt et al., 2014*; *Cheminée et al., 2017*). In general, coastal species with smaller body sizes, such as many Labridae, Blenniidae and Gobiidae, complete their entire life cycle within this habitat. Several other, larger, commercial species, such as various Sparid species e.g., *Pagelus spp* or *Dentex dentex*, often only use it as a nursery habitat, with the larger juvenile and adult life stages moving further offshore. Settlement of juvenile fishes in rocky littoral habitats occurs throughout the year but most species have a settlement peak between early spring and late summer (*García-Rubies & Macpherson, 1995*; *Biagi, Gambaccini & Zazzetta, 1998*; *Bussotti & Guidetti, 2011*). The survival of recently settled juveniles within a habitat, through to their recruitment to the adult population, depends to a large extent on the environmental conditions encountered at the site of settlement (*Beck et al., 2001*). Intolerance to physical extremes, starvation and predation are among the major causes of juvenile mortality in these habitats (*Sogard, 1997*; *Guidetti, 2001*; *Thiriet et al., 2016*; *Cuadros et al., 2018*). High nursery value is thus conferred through a combination of factors that provide adequate physical conditions, refuge space and a sufficient food supply. Highly structured habitats are thought to provide both, shelter and an abundance of food, that may facilitate juvenile survival and growth and thus contribute to overall production and population stability (*Cheminée et al., 2016*; *Dahlgren & Eggleston, 2014*; *Leslie et al., 2017*; *Parsons et al., 2015*;

*Scharf, Manderson & Fabrizio, 2006*). Although the importance of habitat complexity is well documented, it should be noted that it is likely to be context-dependent related to spatial scale and species-specific habitat requirements. For example, highly complex habitats have also been reported to negatively affect juvenile fish, by reducing feeding efficiency (*Tátrai & Herzig, 1995*) and increasing mortality through the harbouring of ambush predators (*Canion & Heck, 2009*).

Nursery habitats that hold large numbers of juvenile fish are important to ensure future recruitment into adult populations. They also contribute to the maintenance of essential food web links and energy transfer processes within many coastal systems (*Gee, 1989*; *Schückel et al., 2013*; *Tito de Morais & Bodiou, 1984*), consuming the micro-fauna (mesograzers e.g., micro crustaceans such as Harpacticoids, Isopods and Amphipods) associated with macroalgae and seagrasses (*Bologna & Heck, 2002*; *Jaschinski & Sommer, 2008*; *Vázquez-Luis, Sanchez-Jerez & Bayle-Sempere, 2008*). Macrophytes often have morphological or chemical deterrents that inhibit direct herbivory by larger consumers, such as fish and urchins (*Cruz-Rivera & Villareal, 2006*). Therefore, meso-grazers that can directly utilize macrophytes are thought to represent an important link in the energy transfer between macroalgae and fish (*Lewis & Anderson, 2000*; *Duffy & Hay, 2000*). Many juveniles are also, in turn, consumed by larger higher trophic level predators (*Doherty & Sale, 1986*; *Sogard, 1997*), thus representing the intermediate stage of energy transfer from primary producers to higher trophic level fish.

Perennial algae species belonging to the genus *Cystoseira* (Fucales, Phaeophyceae) are thought to represent habitats of high quality for juvenile coastal fish in the Mediterranean (*Ruitton, Francour & Boudouresque, 2000*; *Sala et al., 2012*; *Cheminée et al., 2013*). *Cystoseira* are macrophytes with a tree-like morphology that can form dense meadows over rocky substrates, aggregations of which are often referred to as *Cystoseira* forests. This habitat generally shows high primary productivity, is morphologically highly complex and harbours a diverse invertebrate and fish fauna. Therefore, *Cystoseira* forests, in addition to the seagrasses, have been recognised as a central foundation species (as defined by *Dayton, 1972*) of the euphotic zone in the Mediterranean (*Irving et al., 2009*; *Vergés, Alcoverro & Ballesteros, 2009*). Past declines of subtidal *Cystoseira* forests (*Thibaut et al., 2005*; *Sala et al., 2012*) may have had a negative effect on the fish nursery value of coastal rocky reef habitats in many regions of the Mediterranean. Nevertheless, the role and importance of other macroalgae-dominated habitats for juvenile fish has not been fully explored and these could also provide adequate shelter and food for juvenile fish. Thus far, the services or functions that littoral macroalgae may provide to juvenile fish, i.e., shelter and food, have been suggested and only partially tested and quantified scientifically (*Sala & Ballesteros, 1997*; *Ruitton, Francour & Boudouresque, 2000*; *Cheminée et al., 2013*). Thus, overall, we know little about the real habitat value of the littoral macroalgae and the consequences their loss has on the functioning of coastal ecosystems as fish nursery habitats. Furthermore, we are still unaware if there may be other macroalgae habitats, equivalent to *Cystoseira* forests, that may provide a similar function and compensate for their loss. Such knowledge will be of importance for the development of management and conservation strategies against the backdrop of the currently observed acute environmental

change experienced in coastal rocky shores in the Mediterranean, due to chronic pollution, urbanisation, fishing and the introduction of non-native macroalgae species (*Sala et al., 2012*) and herbivorous fish (*Sala et al., 2011*; *Vergés et al., 2014a*) that have the potential to rapidly modify coastal habitats.

The present study aims to quantify the importance of the Mediterranean littoral habitats of varying algal morphotype composition, with respect to its functions for commonly occurring juvenile fish. In particular, the study focused on measuring habitat parameters related to the key habitat requirements for juveniles, i.e., shelter and food. Prey availability in macroalgae has generally been inferred by past studies on nursery habitat quality in the Mediterranean (*Cheminée et al., 2013*; *Cheminée et al., 2017*), but thus far has not been quantified for different algae morphotypes. The study examined the relationship between juvenile density and physical and biological habitat parameters, over two distinct settlement periods (Spring and late Summer) and geographical areas (Mallorca and Menorca). Sampling on these two neighbouring islands was conducted to investigate the generality of any findings within the context of the Balearic Islands. The algae habitats sampled included areas dominated by *Cystoseira* forest and therefore provided the opportunity to contrast this habitat type, with respect to its nursery value, with habitats dominated by other macroalgae. Against the background of rapidly changing algae habitats in the Mediterranean this study aimed to contribute in understanding if a loss of *Cystoseira* habitats will disproportionally affect the nursery function of littoral habitats or if other algae could provide similar functions.

## METHODS

### Fish study species

The study focused on three common and abundant fish species of littoral rocky reefs in the Mediterranean: *Diplodus vulgaris* (common two-banded seabream), *Symphodus ocellatus* (ocellated wrasse), *Coris julis* (Mediterranean rainbow wrasse). These species were chosen as they occurred at sufficiently high abundances to engage with the posed research questions. Settlement of the two-banded seabream, occurs in very shallow rocky coastal areas and seagrass meadows but are thought to migrate quickly to deeper waters following settlement (*Harmelin-Vivien, Harmelin & Leboulleux, 1995*). Settlement peaks for this species have also been reported to occur in two settlement pulses, one in early November and the other in January-through to March (*Vigliolat et al., 1998*; *García-Rubies & Macpherson, 1995*; *Harmelin-Vivien et al., 1985*). *Biagi, Gambaccini & Zazzetta (1998)* describes the settlement periods to occur in December–January and March. Within the Balearic Islands, the authors observed the presence of recently settled juveniles in early spring (April–May).

The settlement period of the two wrasse species has been reported to occur during late summer between July to August (*García-Rubies & Macpherson, 1995*; *Biagi, Gambaccini & Zazzetta, 1998*; *Bussotti & Guidetti, 2011*) while *Raventós (2006)* reports settlement of *S. ocellatus* slightly earlier from June to mid-July. Within the present study, the presence of recently settled juveniles was observed at the end of August by the authors. The two species of wrasse have been reported to settle predominantly in rocky habitats with high algal

cover (*García-Rubies & Macpherson, 1995*; *Biagi, Gambaccini & Zazzetta, 1998*; *Bussotti & Guidetti, 2011*; *Félix-Hackradt et al., 2014*). Juveniles of *D. vulgaris* have been reported to feed predominantly on micro-crustaceans (*Altin et al., 2015*). Adults of *S. ocellatus* feed predominantly on small crustaceans and molluscs with a suspected tendency towards herbivory (*Kabasakal, 2001*) and adult *C. julis* feed on gastropods, sea urchins and small crustaceans (*Sinopoli et al., 2016*). Presently there is no information on the diet of the juvenile stages of these two wrasse species.

## Study area and sampling design

Littoral habitats were studied in the Balearic Archipelago, in the western Mediterranean. Surveys were conducted in early spring (April–May) and late summer (August–September) of 2014, on the islands of Mallorca and Menorca. Sampling in Mallorca took place around the northwestern part, while sampling in Menorca was primarily conducted around the western part (see Fig. 1). Sampling at each island and during each period was conducted in 10 consecutive day surveys. The sampling stations comprised rocky reef zones with a depth of 7–10 m. The same depth range was chosen not to introduce any depth related confounding effects. Furthermore, at this depth we encountered extensive algae communities and juvenile fish. From an operational point of view, this depth allowed for comfortable diving and sufficiently long bottom time for surveying. Stations were haphazardly chosen to cover a variety of habitats, including different physical (geomorphology) and biological (algae cover) properties. Eight stations were sampled per sampling period and island.

In Mallorca the linear distance between consecutive stations was on average 4 km and the survey area extended over a coastal strip of approximately 55 km. In Menorca, the distance between consecutive stations was 7 km on average and the survey area covered a coastal strip of approximately 75 km. Some of the survey sites were sampled both in spring and late summer, while other sites were only sampled within one season. The overall number of samples was however balanced between the two seasons and islands (see above).

## Fish and habitat census

At each sampling station the fish community and the habitat were assessed by diver underwater visual census. The fish assemblage was assessed in 6 replicates of 15 × 2 m transects with a horizontal gap between each of at least 20 m. All fish sighted along the transect were recorded and their size was estimated to the nearest cm (*Harmelin-Vivien et al., 1985*). Benthic composition along the transects was characterized using the following categories: sand, pebbles and gravel, seagrass and rock. The habitat complexity of the substrate was also assessed, using a five-point scale index for rugosity, presence of different sized boulders and amount of refuge spaces (see Appendix S1 for field sampling protocol).

Algae cover, composition and height was measured in three haphazardly placed 50 × 50 cm quadrats along the transect. Quadrats were further subdivided into 25, 10 × 10 cm squares using nylon string. These sub-squares were used to estimate the % cover of 8 algae morphotypes (Fig. 2) and unvegetated barren patches. If a morphotype occurred in a sub-square, the square was counted. For each morphotype, the total number of squares

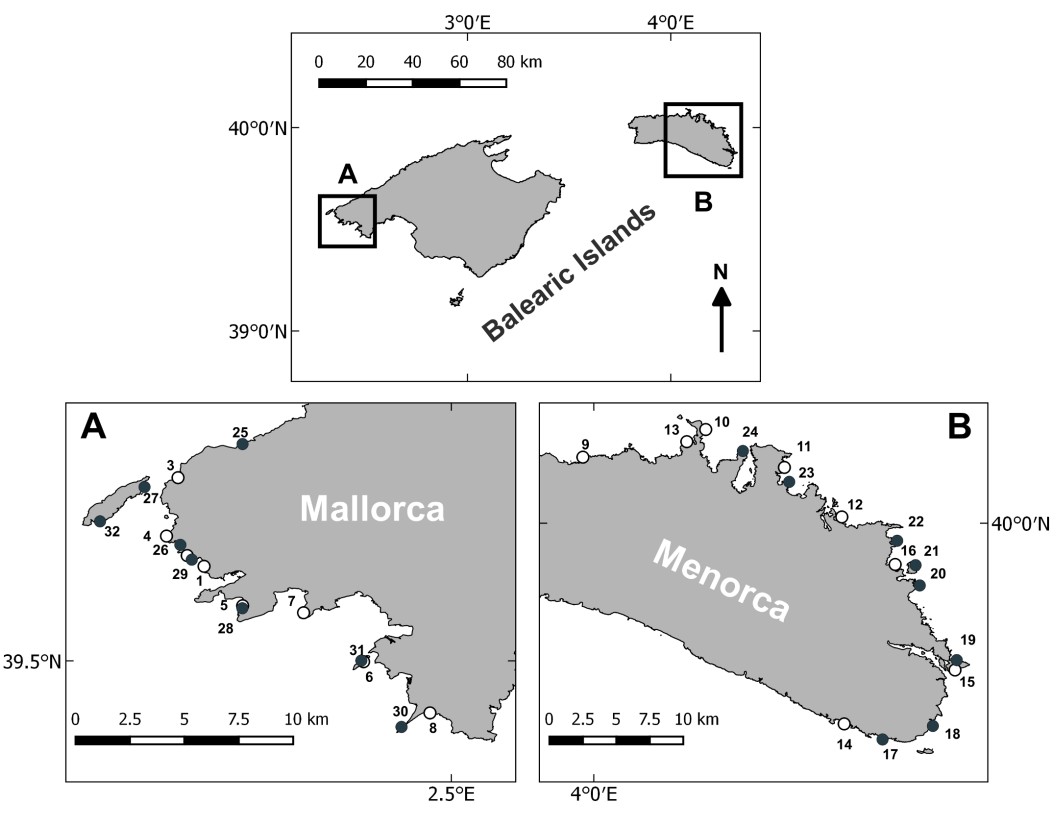

**Figure 1 Survey area and sampling locations.** Sampling stations surveyed in Mallorca (A) and Menorca (B). White circles denote stations sampled in spring and black circles denote stations sampled during the summer.

it occurred in was recorded, as well as their average height in cm form the rock surface. Algae morphotype cover in $m^2$ within transects was calculated by taking the $m^2$ recoded for rocky reef habitats overgrown by algae (as opposed to being occupied by sand or pebble substratum) and this area was dividing proportionally according to the % cover of each morphotype obtained by the quadrate samples. The following algae morphotypes represented by various related species that are commonly encountered in the rocky littoral zone were used (Fig. 2): erect tree like **1-ET** (*Cystoseira* spp.), soft leaf like **2-SL** (*Dictyopteris polypoides* and *Dictyota* spp.), filamentous **3-FI** (*Dictyota dichotoma* var. intricate, Dictyota spp. and occasional *Hincksia* spp.), tubular **4-TU** (*Cladostephus spongiosus*), plumose **5-PL** (*Asparagopsis* spp.), bulbous tree like **6-BT** (*Halopteris* spp.), leathery bands **7-LB** (*Padina pavonica*), turf forming **8-TF** (dominated by Corallinaceae such as *Haliption vigatum*). For data from dive transect see Dataset S1.

## Sampling of algae associated fauna and prey communities

To determine prey availability for juvenile fish, the associated fauna of dominant macroalgae was sampled for each station. At each site, the three most dominant algae types were sampled by collecting $6 \times 113$ $cm^2$ samples. A circular tube of 5 cm height and 11.5 cm diameter was used to define the surface area for algae cuttings (0.01 $m^2$). A 0.55 μm meshed sampling

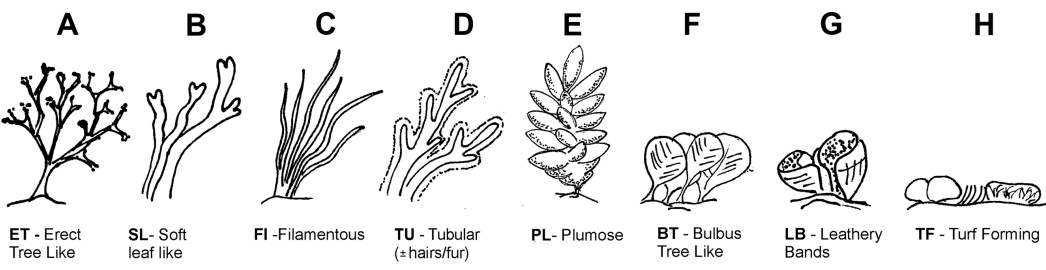

**Figure 2  Algae morphotype.** Algae morphotypes surveyed along dive transects corresponded to different species complexes with similar morphology (A) Erect tree like (*Cystoseira* spp.), (B) soft leaf like (*Dictyopteris* spp. and *Dictyota* spp.),(C) filamentous *Dictyota dichotoma* var. *intricate*, *Dictyota* spp. and occasional *Hincksia* spp.), (D) tubular (*Cladostephus spongiosus*), (E) plumose (*Asparagopsis* spp.), (F) bulbous tree like (*Halopteris* spp.), (G) leathery bands (*Padina pavonica*),(H) turf forming (dominated by Corallinaceae such as *Haliption vigatum*). Copyright H. Hinz.

bag was draped over the piece of tube to collect algae cuttings and to retain associated fauna. The samples were transferred to plastic bags and stored frozen. Once defrosted, the samples were pooled and rinsed into a receptacle using fresh water. The algae were dried with kitchen paper and the wet weight recorded from each station. The water from the receptacle was passed through a 300 μm mesh sieve and any retained fauna was processed. All large fauna (>5 mm) were preserved in 70% ethanol. Using a Folsom plankton sample splitter, the remaining sample, was split up to 4 times depending on its overall volume. The final split fraction of the sample was re-sieved over a 300 μm mesh and preserved in 70% ethanol. The number of sample splits was recorded for each faunal sample to later enable calculation of faunal abundance and biomass of the whole sample.

All fauna retained from the samples was sorted into large taxonomic groups (see Appendix S2 for a full list of groups). A photograph was taken of each taxonomic group using a standard digital camera (Canon powershot G7; Canon, Tokyo, Japan) for the large fraction and a digital camera (EC3) attached to a Leica stereo microscope (MZ16; Leica, Wetzlar, Germany) for the small fraction. The photograph served to measure the size of the organisms and to quantify their abundance. Image analysis software, ImageJ (*Rueden et al., 2017*), was used to measure the size of all organisms using the polyline measurement tool. The pixel length of each measurement line was converted to μm using values attained from calibration images. Length measurements were furthermore converted to biomass using published length-weight relationships of respective taxonomic groups (see Appendix S2 and Dataset S2 for more details on the size-mass conversion and Dataset S3 for data on associated invertebrate fauna size measurements).

Since not all invertebrates collected can be consumed, by juvenile fish, due to their size, stomach content analyses were carried out on a subset of fish. This allowed to determine the taxonomic composition and size of the fauna consumed to more accurately estimate prey densities within algae at each station. Approximately 50 fish of each species were used. These were selected at random, from different stations at which fish samples were available (see protocol of fish sampling below). The taxonomic group and size of taxa were determined from photographs in the same manner as for the determination of algae

associated fauna. Only prey items that had an intact body outline were measured. For data of prey sizes found in stomachs see S6.

To determine the upper size limit of suitable prey for juveniles of each study species (those below 60 mm total length) we used quantile regressions analysis (*Cade & Noon, 2003*). Only significant upper quantiles were considered, starting from the 95th quantile moving down in steps of 5. Since some invertebrate taxa had an elongated, and others had compact body shape, upper prey size limits for respective prey body shapes were calculated for each fish species. The cutoff point for prey sizes to be considered for the determination of prey availability in algae was the value where the upper quantile regression line intersected with the 60 mm total length of the study species (i.e., set length of a juvenile fish) (*Cade & Noon, 2003*). Furthermore, we only considered prey taxa that cumulatively contributed to at least 90% of the prey taxa found in stomachs thus excluding rare species that were only occasionally or accidentally been ingested. As not all algae morphotypes were sampled in both islands, the data on the abundance and biomass of prey per algae morphotype were therefore presented using pooled data from both islands for the season in which juvenile fish of respective study species occurred i.e., *D. vulgaris* spring and summer for *C. julis* and *S. ocellatus*.

Total abundance and biomass of prey within a transect were estimated using the area cover (m$^2$) by different algae morphotypes multiplied by the abundances and biomass of prey items found (per m$^2$) within respective algae cuttings (see above). Organisms were only considered prey if they were of the relevant taxa and sizes as identified by the stomach content analysis described above.

## Sampling of juvenile fish

Using hand-held nets, a sample of approximately 30 juvenile fish of each of the study species were collected from each of the 32 sampling stations (see Dataset S5 for fish size data). In addition to the stomach content analysis, the fish sample was used to evaluate the size distribution of juvenile fish.

## Wave exposure data

As juvenile fish may experience higher dispersal from habitats with high wave action, and algae may be influenced by physical wave stress, wave exposure was considered in the study (*Spatharis et al., 2011*). Wave stress data for each station was calculated from a dynamical coastal wave model provided by the Coastal Ocean Observing and Forecasting System located in the Balearic Islands (SOCIB) using past real weather events. The mean wave exposure, as well as the wave exposure of the 4 months prior to sampling, was used as an environmental parameter in the analysis of data.

## Statistical analysis
### Algae community analysis over temporal and spatial scales
Algae morphotype cover composition was analyzed using a Principal Component Analysis (PCA) based on Euclidian distance. The resulting ordination was explored through investigating the correlation of algae morphotypes with the two first Principle Component (PC) axes. Furthermore, the loading of each variable was superimposed as arrows over

the ordination. The length of the arrow corresponds to the strength of influence on the ordination, while its orientation provides information on the direction of influence. ANOSIM pairwise comparisons were used to test for significant differences in algae cover composition between the transects sampled between different seasons and islands. Distance based linear models (DistLM) were used to investigate the environmental variables that best explained the observed ordination patterns in algae composition. The following five environmental variables were used: mean depth, slope, rugosity, temperature, wave stress and herbivore density (*Salpa salpa* and sea urchins), and. All multivariate analyses were conducted with the Software Package Primer-E version 6 (*Clarke & Gorley, 2006*).

The height of different algae morphotypes was compared on an island scale using a two-way ANOVA with morphotype and season as main factors. A Post-hoc Tukey-test was used to identify height differences between morphotypes within seasons and the height of the same morphotype between seasons. Height data was log-transformed prior to analysis to meet model assumptions regarding normality and homogeneity of variance.

## Potential prey availability in different algae morphotypes for juvenile fish

Potential prey availability was calculated considering prey sizes, taxonomic groups and the season relevant to the juveniles of the respective fish species (<60 mm). Potential prey availability was compared between morphotypes using a one-way ANOVA and post-hoc Tukey pairwise comparison tests. The data of both islands were pooled to compare all algae morphotypes since for some algae morphotypes insufficient specimens were sampled to be able to consider both islands separately in the analysis. Potential prey density was log-transformed prior to analysis to meet model assumptions regarding normality and homogeneity of variance.

## Relationship between different algae morphotypes and juvenile fish abundances

The relationship between juvenile fish abundance and algae morphotype cover ($m^2$), as well as total prey biomass per transect, was modeled using a Generalized Linear Mixed Model (GLMM) with a negative binomial error distribution and a log-link function. Within the model the station was considered a random factor, to address the dependency structure of dive transect from the same station. The cover by morphotypes, as well as total prey biomass per transect, were considered as fixed factors. All variables introduced to the model were correlated prior to analysis to identify any collinear variables to be removed or pooled prior to modelling. Model selection procedures were adopted whereby non-significant variables were removed until only significant variables were contained in the final model, using the drop-one procedures in R. Models were checked for over dispersion and the residuals were visually examined (*Zuur et al., 2009*; *Zuur, Ieno & Elphick, 2010*).

## RESULTS

### Algae morphotype composition and height across sampling seasons and islands

Algae morphotype composition within sampling areas and season were compared by PCA ordination of the dive transects sampled. The PCA explained 60.5% of the variability of the algae cover data (Fig. 3, Table S1). The ANOSIM pairwise comparisons analysis verified that there were significant differences in composition of morphotypes between all island and season combinations ($p < 0.001$, Fig. 3). The largest differences were found between transects surveyed in summer in Menorca and all other island-season combinations (see Fig. 3 and Table S1). Algae cover was most similar between spring and summer in Mallorca (see Fig. 3). Transects sampled in spring in Menorca were similar to those sampled in Mallorca (see overlap of PCA space Fig. 3 and lower summed Euclidian squared distance Table S2). Overall, seasonal changes in algae cover for both islands were similar for most morphotypes (Table 1). In both islands, the average transects cover of 1-ET (*Cystoseira* spp.) were lower in summer compared to spring as was the cover of 2-SL and 3-FI (*Dictyota* spp. and similar). In contrast, the cover of 7-LB (*P. pavonica*) was higher in both islands during summer as did the occurrence of barren patches (no vegetation over rock) (Table 1). However, the magnitude of seasonal change was more pronounced in Menorca. Furthermore, transects surveyed in summer were significantly different to those sampled in Mallorca (Fig. 3 and Table S2) with higher cover in morphotype 7-LB (*P. pavonica*) and lower turf forming morphotypes 8-TF (Corallinaceae) compared to Mallorca (Fig. 3 and Table 1). Morphotypes 2-SL and 3-FI (*Dictyota* spp. and similar) formed part of many transects in Mallorca in late summer, while these morphotypes were almost absent from transects in Menorca (Fig. 1 and Table 1).

The distance-based linear model (DistLM) relating environmental variables to the PCA ordination of algae cover showed that of the 6 environmental variables considered, three (Temperature, wave stress and rugosity) proved to show significant relationships with the PCA ordination pattern (Fig. 3 and Table S3). The model with the best fit (lowest AIC) containing these three variables had an $r^2$ of 0.2 thus explaining about 20% of the variability. Mean depth, slope and herbivore density (*Salpa salpa* and urchins) were not significantly correlated with the ordination (Table S3). Temperature and wave stress were associated with PC1 as indicated by the horizontal orientation of the eigenvectors in Fig. 3 and thus could be related to the distinct algae cover in Menorca over the summer period. In general, average wave stress was higher ($H_s$ 0.7) and temperature lower (25.8 °C) in Menorca compared to Mallorca during summer surveys ($H_s$ 0.4 and 26.9 °C respectively).

Additional to the surface cover, the height of algae varied between morphotypes, seasons and islands (Fig. 4). The height of morphotypes was higher in Mallorca compared to Menorca. Seasonal changes in the height of cover were, however, relatively consistent over both areas. Overall, *Cystoseira* spp. (1-ET) decreased in height between spring and summer surveys (Fig. 4). In Mallorca *Cystoseira* spp. decreased in height from spring to late summer from an average 11.2 cm to 4.3 cm, while in Menorca it decreased from 7.6 cm to 5.1 cm. These reported changes were statistically significant (Tables S4 and S5). Other morphotypes

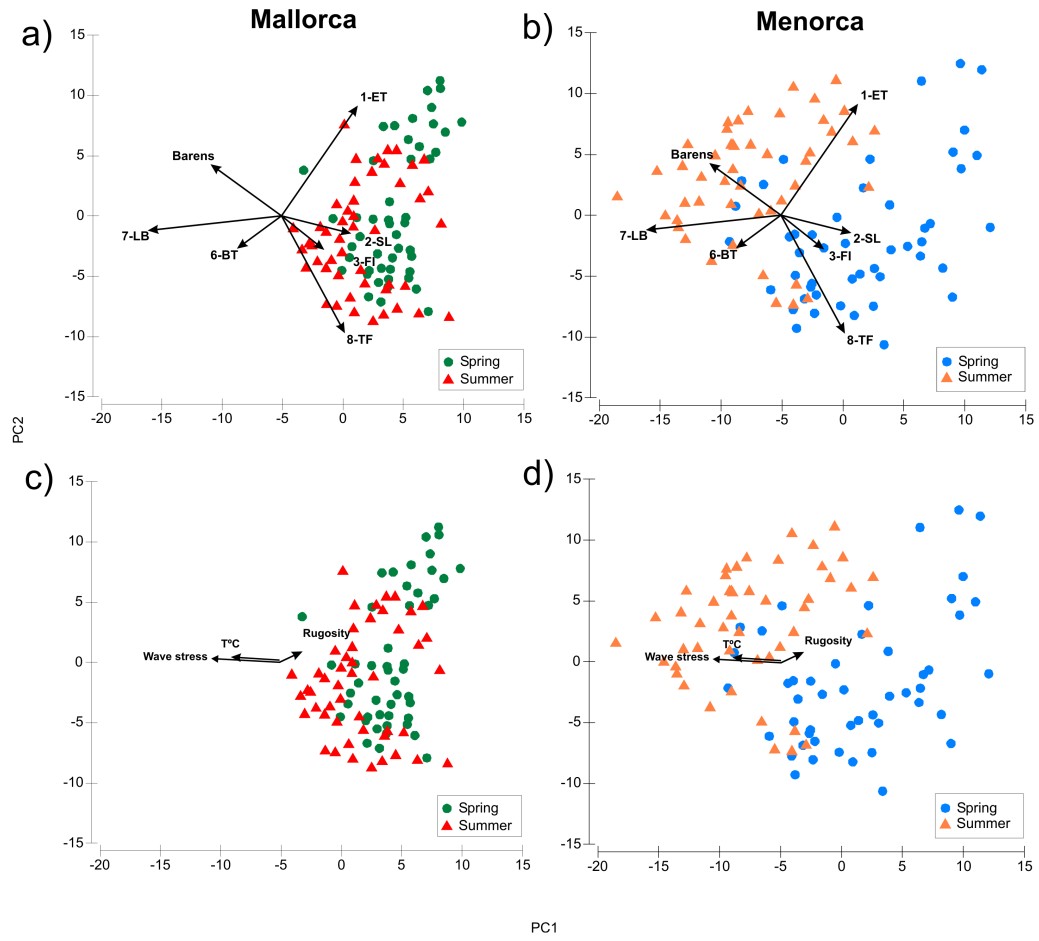

**Figure 3** **Comparison of algae morphotype composition in dive transects sampled during spring and summer in Mallorca and Menorca.** PCA showing the composition of algae morphotype cover over dive transects surveyed in summer and spring (A: Mallorca, B: Menorca), as well as the potential environmental drivers in the variation of algae morphotype cover. (C: Mallorca, D: Menorca).

decreased less in height (2-SL) or approximately maintained their height (3-FI, 6-BT and 8-TF). In summer, *P. pavonica* (7-LB) was the only algae that significantly increased in height in both areas from 4.2 to 6.9 cm in Mallorca and 3.8 to 6.8 in Menorca. During the summer period in Mallorca filamentous morphotype algae (3-FI) were found to be the highest with 10.6 cm, while in Menorca both filamentous (3-FI) and bulbus tree-like (6-BT) were the highest with 7.8 and 7.9 cm respectively.

## Taxonomic composition and size of prey items found in juvenile fish stomachs

Stomach samples of the three-study species showed that all three species shared many prey taxa (see Fig. 5). *D. vulgaris* and *S. occelatus* diets were dominated harpacticoids (both 53%) while stomachs of *C. julis* predominantly by gastropods (50%) and to a lesser extent Harpacticoids (34%). *D. vulgaris* stomach samples were furthermore dominated by
**Table 1 Morphotype cover during spring and summer in Mallorca and Menorca.** Percentage occurence of algae in dive transects and mean algae morphotype cover (m²) in Mallorca and Menorca during spring and summer sampling. Arrows indicate average increases or decreases in algae cover between sampling seasons at respective Islands.

| | Mallorca | | | | | | | Menorca | | | | | | |
|---|---|---|---|---|---|---|---|---|---|---|---|---|---|---|
| Season | Spring | | | | Summer | | | Spring | | | | Summer | | |
| Morphotypes | % occ. T. | Mean | S.D. | | % occ. T. | Mean | S.D. | % occ. T. | Mean | S.D. | | % occ. T. | Mean | S.D. |
| 1-ET | 89.6 | 5.8 | 5.1 | > | 79.2 | 3.0 | 3.7 | 64.6 | 5.6 | 6.0 | > | 60.4 | 4.0 | 4.1 |
| 2-SL | 100.0 | 4.1 | 2.6 | > | 18.8 | 2.6 | 2.6 | 2.1 | 0.2 | 0.8 | > | 87.5 | 0.1 | 0.8 |
| 3-FI | 95.8 | 3.5 | 2.1 | > | 83.3 | 1.9 | 2.9 | 35.4 | 3.5 | 3.3 | > | 56.3 | 0.6 | 1.1 |
| 4-TU | 56.3 | 1.5 | 2.8 | > | 12.5 | 0.4 | 1.4 | 6.3 | 0.0 | 0.1 | > | 16.7 | 0.1 | 0.6 |
| 5-PL | 45.8 | 0.5 | 1.0 | > | 14.6 | 0.0 | 0.0 | 0.0 | 0.1 | 0.2 | > | 0.0 | 0.0 | 0.0 |
| 6-BT | 85.4 | 1.7 | 1.6 | > | 66.7 | 0.4 | 1.2 | 64.6 | 1.9 | 2.6 | < | 27.1 | 2.4 | 3.3 |
| 7-LB | 95.8 | 2.7 | 1.8 | < | 97.9 | 4.0 | 2.7 | 100.0 | 6.8 | 4.0 | < | 91.7 | 12.2 | 5.1 |
| 8-TF | 100.0 | 6.8 | 3.5 | > | 100.0 | 9.4 | 4.0 | 72.9 | 9.9 | 4.5 | > | 100.0 | 3.2 | 3.8 |
| Barens | 37.5 | 0.5 | 1.0 | < | 33.3 | 5.6 | 3.5 | 97.9 | 0.7 | 2.2 | < | 91.7 | 5.8 | 3.7 |

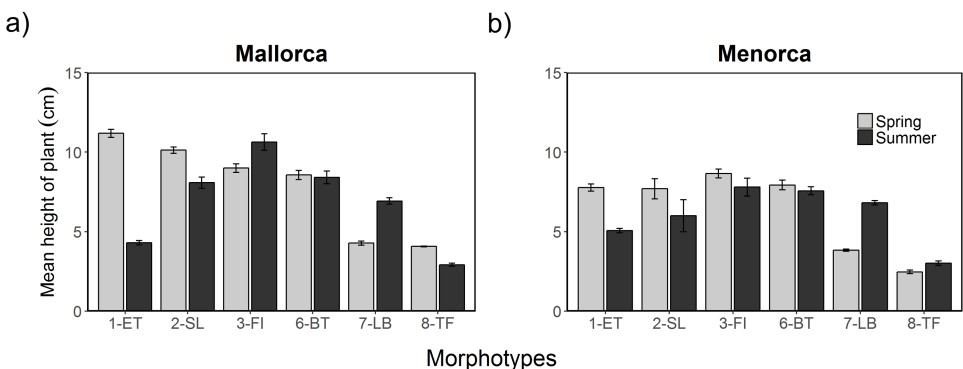

**Figure 4 Height of algae morphotypes during spring and summer in Mallorca and Menorca.** Mean height of algae morphotypes during spring (May 2014) and summer (August 2014) in (A) Mallorca and (B) Menorca measured within 0.5 m² quadrats. Error bars are S.E.

Ostracods (14%), Amphipods (9%) and Gastropods (9%), while *S. occelatus* also contained sea mites (Acari 34%) and Gastropods (15%).

For all species, significant upper quantiles ($p < 0.05$) were found determining the upper limit of prey sizes that were consumed at a certain size (Fig. 5). Comparing the upper limit of prey sizes consumed with increasing size of fish showed that fish below 60 mm, of the three-study species consumed very similar size prey when considering compact body prey such as Harpacticoids or Amphipods. In general prey sizes were between 1.33–1.61 mm. Both *D. vulgaris* and *C. julis* consumed elongated prey consisting mainly of Polychaetes reflected in the % contribution to the diet, 3% and 6% respectively (Fig. 5). Other elongated taxa such, *Diptera* larvae, *Caprellidea* and *Tanaidacea* contributed less to this prey category. Stomachs of *S. occelatus* had too few elongated prey items to perform an upper quantile regression analysis. The size of elongated prey found in *C. julis* were slightly larger for fish at a size of 60 mm compared to *D. vulgaris*. The size elongated taxa consumed was below

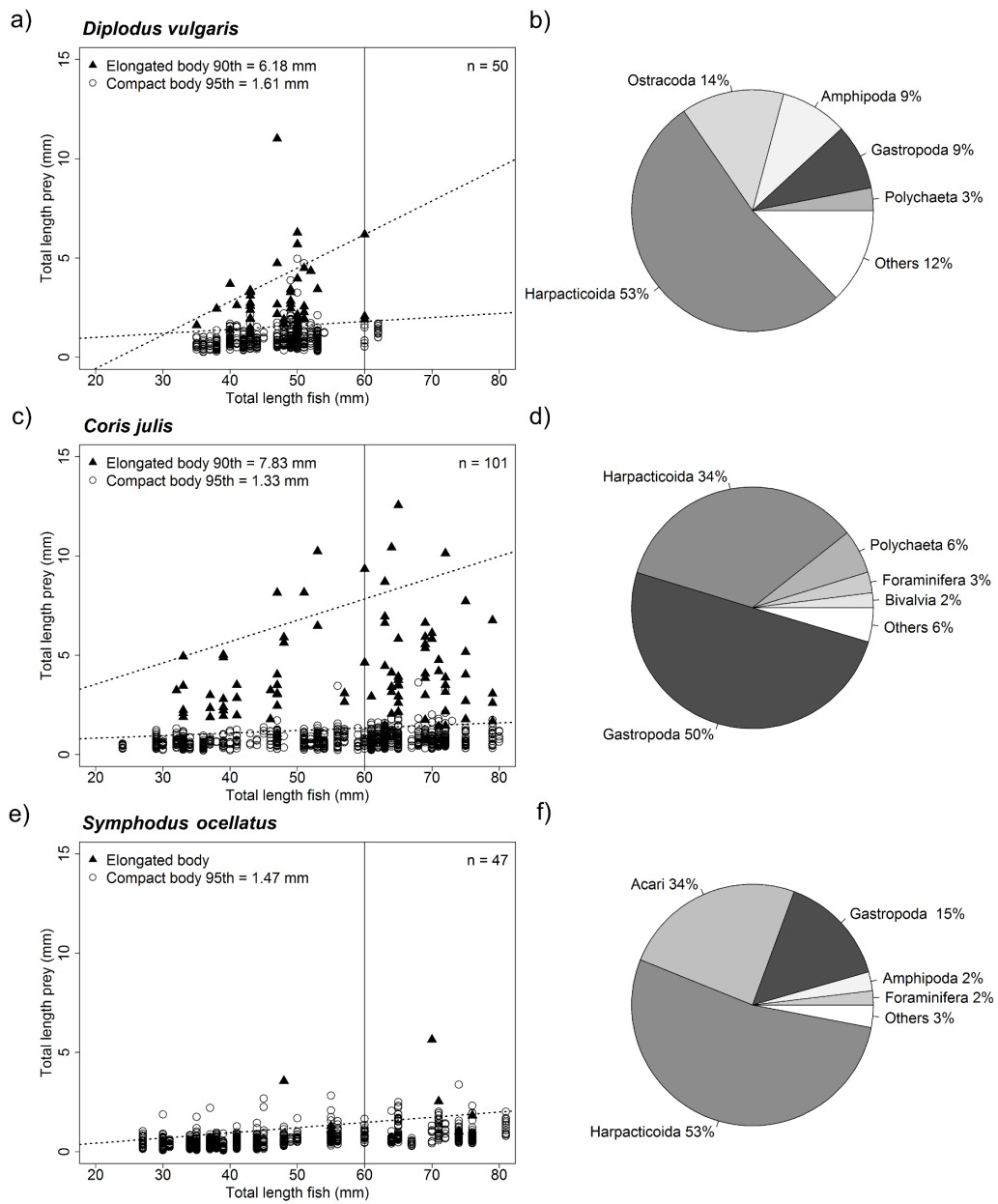

**Figure 5** **Relationship of fish size and prey size as well as overall taxonomic composition of prey ingested.** Graph (A, C and E) showing size distribution of ingested pey for different sized predators. The upper prey size limit for fish of up to 60 mm was determined by quantile regression for both elongated body prey (i.e., Polychaetes, Diptera larvae, Caprellidea and Tanaidacea) and compact body prey. Only significant upper quantiles were considered starting from the 95th quantile moving down in steps of 5 points. The first significant upper quantile was considered the upper limit. Due to the rare occurrence of elongated prey in *Symphodus ocellatus* stomachs no specific upper boundary could be established for elongated prey for this species. Size limits of prey for fish below 60 mm total length for both prey body shapes are given in the graphs (intersect of upper quantile regression with 60 mm vertical line), as well as the number of fish analyzed (*n*). Pie charts (B, D and F) showing the proportional contribution of taxa found in fish stomachs analyzed. The five most dominant taxa are shown, all other taxa were pooled under the category others.

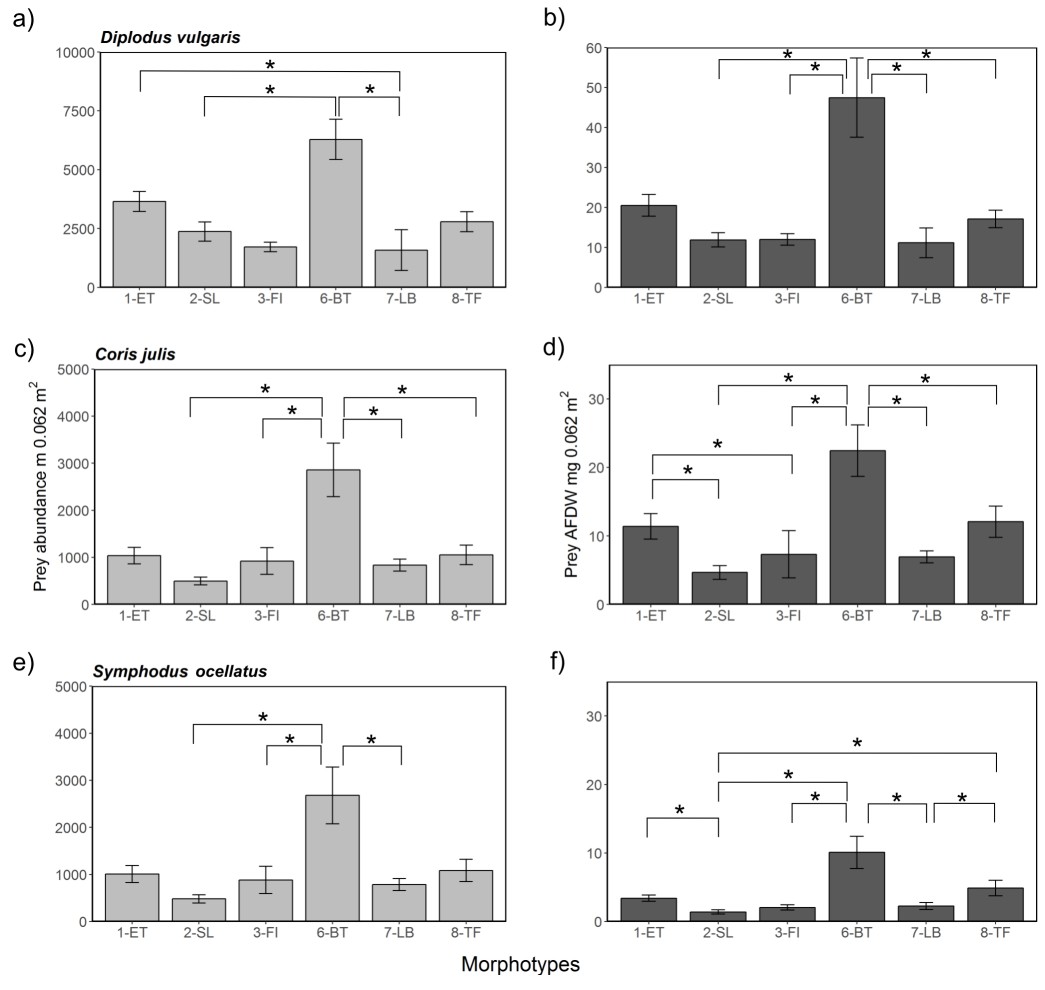

**Figure 6  Prey availability in different algae morphotypes.** Prey availability with respect to abundance (A, C, E) and biomass as AFDW (B, D, F) per 0.067 m² within different algae morphotypes. Significant ($p < 0.05$) pairwise comparisons of from the post-hoc analysis are marked with a bracket and an asterisk.

6.18 mm and 7.83 mm respectively (see Fig. 5). Overall the stomach content reflected well the dominant taxa found in the different algae morphotypes (Table S6).

## Potential prey availability in different algae morphotypes for juvenile fish

Comparing potential prey abundance and biomass by algae morphotype using one-way ANOVA analysis showed that there were significant differences considering the three-study species (Table S7 and Fig. 6). Post-hoc Tukey tests showed that in general 1-ET and 6-BT held significantly higher prey densities and biomass compared to other algae morphotypes (S15 and Fig. 6). Overall, the highest prey densities or biomasses in absolute terms were, irrespective of season and fish species, found in the three structurally complex algae 1-ET, 6-BT and 8-TF, compared to the more structurally simple algae, such as 2-SL, 3-FI and 7-LB. The general pattern between islands and seasons observed in prey abundances and
**Table 2  Number and sizes of juvenile fish observed during spring and summer in Mallorca and Menorca.** Summary table of the total number of fish observed in visual censuses and their sizes in mm, measured from a capture subsample (see 'Method' section). The table also includes the number of other Sympodus species encountered and the number of adults observed over transects by sampling season and island.

| | | Spring | | Summer | |
|---|---|---|---|---|---|
| | | Mallorca | Menorca | Mallorca | Menorca |
| *D. vulagris* | No. of juveniles | 35 | 4 | – | – |
| | Size of juveniles in mm (caught) | 43.8 (3.8) $n = 105$ | 46.1 (5.7) $n = 46$ | | |
| | No. of adults | 188 | 168 | 508 | 229 |
| *C. julis* | No. of juveniles | – | – | 181 | 245 |
| | Size of juveniles in mm (caught) | | | 48.7 (5.2) $n = 19$ | 32.7 (10.3) $n = 168$ |
| | No. of adults | 314 | 373 | 743 | 528 |
| *S. ocellatus* | No. of juveniles | – | – | 486 | 63 |
| | Size of juveniles in mm (caught) | | | 28.1 (14.6) $n = 371$ | 33.3 (9.7) $n = 51$ |
| | No. of adults | 220 | 90 | 186 | 34 |
| | No juveniles other *S. spp.* | | | 48 | 20 |

biomass were reflective of patterns when considering all associated fauna (Fig. S1). Note that overall abundances and biomass of associated fauna were considerably higher within all algae morphotypes in spring compared to summer (Fig. S1).

## Relationship between different algae morphotypes and juvenile fish abundances

The total number of juvenile fish observed over the visual transects surveyed varied considerably between islands and species. For *D. vulgaris* we only encountered 39 fish in 96 transects during the spring period of which 35 were recorded in Mallorca (mean size 43.8 mm Mallorca and 46.1 mm Menorca). In contrast, similar numbers of *C. julis* were recorded at both islands during the summer sampling period see Table 2 (mean size 48.7 mm Mallorca and 32.7 mm Menorca). However, for *S. ocellatus* the number of juveniles varied considerably between islands. In 48 transects we observed 486 fish in Mallorca as opposed to only 63 in Mallorca (mean size 28.1 mm Mallorca and 33.3 mm in Menorca). This island scale difference was also reflected in other *Symphodus* species (see Table 2).

Due to the low abundance of *D. vulgaris* juveniles in visual census transects we did not investigate their relationship with algae cover and total prey biomass within transects using a Generalized Linear Mixed Modelling (GLMM) approach. Furthermore, we restricted this type of analysis to Mallorca for *S. ocellatus* where we had a sufficiently large sample size. *C. julis* was analyzed considering data from both islands.

Prior to the GLMM analysis we correlated the explanatory variables to detect any collinearity. As we found a positive correlation between 2-SL and 3-FI (Pearson correlation coefficient 0.68) and as the two algae morphotypes were structurally and taxonomically similar, their cover was pooled to avoid collinearity.

For *C. julis* neither algae cover, nor the total prey biomass had a significant effect on juvenile abundances. For *S. ocellatus* the best model including only the significant variables contained the algae morphotypes 2-SL+3-FI and 7-LB. While 2-SL+3-FI had a positive effect on fish abundances 7-LB had a negative effect (Table 3 and Fig. 7).

**Table 3  Relationship of *S. occelatus juvenile* density with algae morphotypes.** Summary table of the total number of fish observed in visual censuses and their sizes in mm measured from a capture subsample (see 'Method' section). The table also includes the number of other Sympodus species encountered and the number of adults observed over transects by sampling season and island.

| GLMM Species | Conditional model: | Estimate | Std.Error | $z$-value | $p$-value | |
|---|---|---|---|---|---|---|
| *S. ocellatus* | (Intercept) | 2.10 | 0.28 | 7.37 | 0.0000 | *** |
| | 2-SL + 3-FI | 0.12 | 0.03 | 3.72 | 0.0002 | *** |
| | 7-LB | −0.15 | 0.06 | −2.38 | 0.0172 | * |

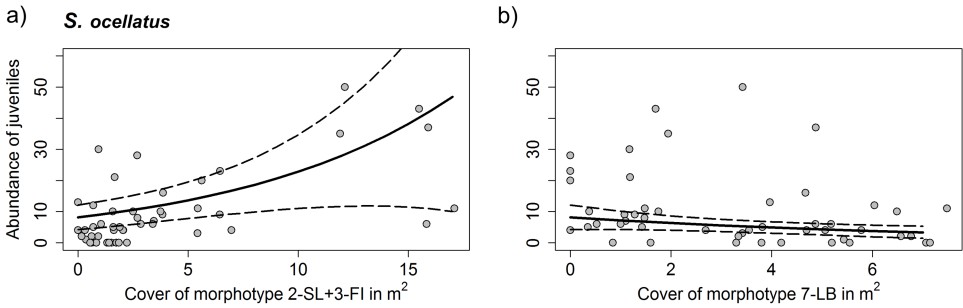

**Figure 7  Density of *S. ocellatus* with different algae morphotype cover.** Results of the generalized mixed variable intercept model (A) morphotypes 2-SL+3-FI (Dictyotales) and (B) morphotype 7-LB (*P. pavonica*). Solid line represents predicted juvenile abundances of the GLMM model and dashed lines the 95% C. I. Data points represent individual transects. Detailed summary of results presented in Table 3.

# DISCUSSION

High quality habitats for juvenile coastal fish have generally been characterized by both, high sheltering opportunities and food for the resident juvenile fish (*Cheminée et al., 2013*; *Cheminée et al., 2017*). In our study, we however observed that the occurrence of juvenile fish in late summer appeared to coincided with a period of less optimal habitat conditions. Our results showed strong seasonal variations in algae cover between sampling seasons. Algae morphotype cover and height had changed considerably on both islands between the two sampling events that coincided with the two major settlement peaks of juvenile fish documented for the Mediterranean (*García-Rubies & Macpherson, 1995*; *Biagi, Gambaccini & Zazzetta, 1998*; *Bussotti & Guidetti, 2011*). During the settlement period of *D. vulgaris* the extent and height of the forest forming algae *Cystoseira* spp. was considerably greater compared to the summer period in which *S. ocellatus* and *C. julis* settled. Similarly, most other algae decreased in cover and to some extend in height except for *P. pavonica*, which increased both in cover and height during this period. Concurrent with these cessation patterns of algae, the associated fauna was also found to undergo a sharp seasonal decline in abundance and biomass, from spring to summer. However, this was not restricted to algae that decreased in height but was a consistent pattern which was also found to affect algae that increased in height, such as *P. pavonica*. Thus, when comparing overall prey availability per unit area between fish species, there were considerably more prey available

in algae during the spring period at *D. vulgaris* settlement, compared to prey available for the other species during the late summer period. These seasonal patterns between algae and associated fauna have also been demonstrated by other authors (e.g., *Guerra-García et al., 2011*). An apparent mismatch exists between the period of high shelter and food provision by habitats and the settlement and growth period of juvenile fish. Summer settlement of juveniles results from spawning during early spring and the mismatch may be caused by evolutionary processes favoring the match of earlier larvae stages with planktonic production processes, ensuring higher larval survival. Certainly, for *C. julis* there appear to be strong links between primary production processes and the amount of larval settlement (*Fontes et al., 2016*), which may also apply to other fish species with summer settlement peaks in the Mediterranean. This mismatch, therefore, does not suggest that shelter and food are not important for juveniles, but that they do not appear to have a strong selective influence on the timing of spawning. The decrease of shelter and food provisioning from spring to summer shows that coastal habitats and the services they provide for juveniles are not a constant and that they can undergo notable changes in a relatively short amount of time. With this in mid, it is plausible that climate change may further increase the mismatch between algae habitat provisioning properties and summer settlers, if higher temperatures increase the speed of seasonal cycles causing earlier cessation in algae communities. Little is currently known about the influence of climate change on the timing of seasonality in macroalgae. Several studies argue that warming can affect the phenology and reproduction of macroalgae (*Kraufvelin et al., 2012*; *Andrews, Bennett & Wernberg, 2014*), thus suggesting that early cessation processes could be triggered by seasonally early or extreme warming events. For example, in the Baltic, increased seawater temperature and light during early springs accelerate receptacle growth of *Fucus vesiculosus*, causing earlier reproduction (*Kraufvelin et al., 2012*). As cessation processes generally follow reproduction in many marco-algae species (*Liu et al., 2017*), climate change has the potential to shorten macroalgae cycles potentially causing a mismatch between the provision of shelter and food for juvenile fish (*Durant et al., 2007*). Both the timing of spawning and the rate of larvae and juvenile development may, however, also be affected by temperature (*Pankhurst & Munday, 2011*). Therefore, it is also conceivable that both juvenile fish and habitat may be affected synchronously. The precise effect of temperature on reproduction and larvae development of fish is, however, complex and highly species-specific. Thus, for example warming may advance or delay the maturation and ovation depending on species specific thermal endocrine triggers (*Pankhurst & Munday, 2011*). Climate induced mismatches between fish and their food resources have been reported for pelagic fish, illustrating that climate warming may not affect larval development and prey production equally (*Gröger, Hinrichsen & Polte, 2014*; *Illing et al., 2018*). Whether the match/mismatch hypothesis, i.e., the synchronized/desynchronized timing of seasonal activities, as described by Cushing (*1969*, *1990*), is important with respect to algae habitats provisions for juvenile fish is currently unknown and requires further scientific attention.

When considering the potential prey availability in different macroalgae morphotypes, we found that structurally more complex algae contained more prey, in particular, algae such as *Halopteris* spp. and *Cystorceria* spp., but also turf forming *Corallinaceae* such as
*Haliption vigatum*, that have a complex internal structure. In contrast, Dictyotales, with flat leafs or filamentous algal species, contained fewer prey. This pattern of distribution of prey amongst algae was persistent during both settlement periods. Whether algae containing higher prey biomasses also provide higher quality habitats is unclear. The term potential prey availability was used throughout as we did not examine whether this prey was readily available to the juvenile fish. It is possible that, due to the complex structure of some algae, feeding efficiency of juveniles may be decreased or less optimal (*Tátrai & Herzig, 1995*). Thus, the higher amounts of associated invertebrate fauna in morphologically complex algae could reflect reduced predation by fish. Equally, lower biomass of potential prey within less complex algae morphotypes could be the result of increased feeding on less protected and thus easier to feed upon prey. As a result, it is possible that juveniles are responsible for structuring the associated algal fauna. Further research into juvenile feeding behavior will be required to explain observed prey distributions among algae morphotypes.

Within our study, no tangible indication appeared within the data that juvenile fish densities are strongly linked to prey densities. For example, *S. ocellatus*, was associated to algae with the least prey. It is likely that food availability is of secondary importance to fish abundance, but it may, nevertheless, affect juvenile body condition and growth rate and thus their long term-survival (*Lloret et al., 2002*; *Lloret et al., 2012*). To assess and compare nursery qualities of habitats and move away from using solely abundance estimates for habitat suitability, these parameters should be considered by future studies.

Analyzing the association of juvenile fish abundance and algae morphotype cover, we discovered that, while for *C. julis* there was no clear association to any particular algae morphotype, *S. ocellatus* had a positive association with Dictyotales and a negative association with *P. pavonica*. No analyses were carried out for *D. vulgaris*, as too few individuals were observed within transects. The absence of a clear relationship of *C. julis* with algae morphotype appears consistent with findings by *Cheminée et al. (2017)*, who reported an association of this species with sparser algae cover. However, we found no clear relationship between the abundance of *S. ocellatus* and *Cystoseria* contrary to the patterns observed by other studies (e.g., *Cheminée et al., 2013*; *Cheminée et al., 2017*; *Thiriet et al., 2016*). These studies stress the importance of *Cystoseira* forests as a prime juvenile habitat for *Symphodus* spp., due to their structure. These discrepancies could be due to different factors such as: (i) the different UVC methodlogy used in the different studies (15*2 m transects this study, 1 m$^2$ point-count (*Cheminée et al., 2013*; *Cheminée et al., 2017*) and 9-m$^2$ stationary-point snapshot count (*Thiriet et al., 2016*)), which can have an effect on the total amount of fish counted; (ii) different geographical and temporal settlement peaks of local populations, which can produce a mismatch between the time of the study and the time of the settlement in each study area and year of sampling (iii) seasonal and annual differences in the development of height and cover of *Cystoserira*. Within our study Dictyotales provided considerably higher cover, compared to *Cystoseira*, and may have therefore been more important in providing shelter to *Symphodus* spp. The low numbers of individuals of this species in Menorca were not only restricted to juveniles but also fewer adults were observed, possibly indicating that habitats surveyed in this island were altogether less suitable for this species. Apart from potentially having provided

better sheltering opportunities, Dictoytales were observed to be an important nest building material for male *S. occelatus*, during spring in Mallorca. Many labridae species build nests in which the male protects and broods the fertilized eggs (*Raventós, 2006*). Thus, it is possible that the higher abundances of juveniles observed in Dictoytales dominated habitats may be related to a higher number or quality of nests, subsequently providing a locally larger numbers of settlers in summer. However, a study on the related species *Symphodus roissali* found no significant relationship between the indirect measure of successful nests (larval output) and the number of recruits within a three year period (*Raventos, 2009*). It remains unclear if the shelter, or better nesting conditions, were responsible for higher abundances of *S. ocellatus* in Dictoytales in our study.

Overall, more effort needs to be made to study the behavioral relationship and interaction of fish with their habitats. This and other studies focused on measuring the percentage cover of algae for shelter and food provisioning. Shelter, however, may be used in different configurations. For example, fish may use banks/patches of algae to hide from predators behind or hide within the algae matrix. Hiding behind would require high cover but in a fragmented format, with open spaces, whereas hiding within would require a more continuous cover. In our study, while catching juvenile fish, we observed that *S. ocellatus* hid within the algae matrix, while *D. vulgaris* avoided entering any algae or *Posidonia* patches but instead used these as a parkour of obstacles to outpace and loose human pursuers. In the case of *D. vulgaris,* a continuous dense algae forest cover, may due to this behavior, not represent an ideal habitat. This type of behavioral shelter use may explain why no trends were observed for this species and shows that future studies also needs to consider the spatial configuration of cover depending on the behavioral needs of a species.

## Conclusion and recommendations for future studies

The results of this study show that in many ways we still have a relatively rudimentary understanding of what represents a good nursery habitat for juvenile fish. Algae species such as *Cystoseira* that appear to have high importance in one area and season for a specific species may not necessarily have the same importance in another area or season. This may be due to the context dependent nature of juvenile occurrence and the habitat composition at the time of the study. The results suggest that fish can adapt to local situations and use algae morphotypes/species that offer the required function at the time. While at the time of our study we detected an association of *S. ocellatus* juveniles (3–4 cm) with Dictoytales these results do not implicate that *Cystoseira* habitats are not important for juvenile fish. Settlers that arrived in July-early August may well have used *Cystoseira* to shelter within the first two months of their lives. Equally likely, those new settlers of other species arriving in the area during the spring, when the *Cystoseira* forest are better developed with higher cover and canopy height may have a preference for this algae over others. While providing new insights, the present study is not able to evaluate the importance of *Cystoseira* habitats as a nursery habitat with certainty and as such the effect of the continuing loss of this habitat. Considerably more focused research will be required to address this question.

Abundance of juvenile fish may vary considerably in time, at any one location, due to natural stochastic processes, causing a high degree of variability. Scenarios are imaginable

in which, due to a strong settlement pulse and fishery-depleted predators, there are many juveniles in a habitat. However, because of the lack of food provided by the habitat, fish may have low long-term survival potential (e.g., *Macpherson et al., 1997*; *Planes et al., 1999*; *Cuadros et al., 2018*). Using abundance indices alone as a tool to assess habitat quality has limitations, as these do not consider the effect of different habitat conditions on the individual fish, i.e., how a habitat affects parameters such e.g., growth and survival potential. Thus, when combined with abundance estimates, methodologies that assess fish condition, growth performance and health status may provide a more accurate picture of the prevailing conditions for fish living at a certain location. Habitat quality for juveniles should not be measured on the level of "settlement success" alone i.e., the maximum number of recently settled individuals, which may be largely a function of stochastic hydrodynamic processes (*Beck et al., 2001*). A better measure of habitat quality would be to estimate the survival rate of newly settled fish to reach a certain arbitrary size in good condition. However, measuring this type of "recruitment success" would be time consuming and financially costly as it would require intensive temporal sampling of juveniles from their settlement phase to their recruitment to the adult population. To increase our understanding of what constitutes high quality nursery habitats, it is important to have viable and cost-effective sampling solutions. Therefore, future studies should aim to incorporate methods and develop indicators that link both abundance estimates and population fitness parameters of juvenile fish at an agreed, predefined size range (by species)to evaluate habitat quality.

## ACKNOWLEDGEMENTS

This study was logistically supported by the the *Laboratorio de Investigaciones Marinas y Acuicultura* (LIMIA) and we would like to thank Elena Pastor and Amalia Grau for their help and commitment. We would like to thank the Balearic Islands Coastal Ocean Observing and Forecasting System (SOCIB) for the provisioning of wave stress data from their forecasting models. Here particular thanks to Amaya Álvarez. We also would like to thank Grace Niamh Tomlinson and Ada Barbanera for assisting in the laboratory work. We also would like to thank Emil Ólafsson for earlier reflections on the study.

### Funding

This research received funding through the FP7-People IEF—Marie-Curie Action project LINKFISH (299552) "Investigating the link between sub-littoral algae habitats and fish communities in the Mediterranean Sea". During part of the write-up of this work, Hilmar Hinz was supported by the Ramón y Cajal Fellowship (grant by the Ministerio de Economía y Competitividad de España and the Conselleria d'Educacio, Cultura i Universitats Comunidad Autónoma de las Islas Baleares) and Andrew F. Johnson was supported by NSF grant DEB-1632648 (2016-18). The funders had no role in study design, data collection and analysis, decision to publish, or preparation of the manuscript.

## Grant Disclosures

The following grant information was disclosed by the authors:
FP7-People IEF—Marie-Curie Action project LINKFISH: 299552.
Ramón y Cajal Fellowship.
Ministerio de Economía y Competitividad de España and the Conselleria d'Educacio.
Cultura i Universitats Comunidad Autońoma de las Islas Baleares.
NSF: DEB-1632648 (2016-18).

## Competing Interests

The authors declare there are no competing interests. Andrew F. Johnson is an employee of MarFishEco.

## Author Contributions

- Hilmar Hinz conceived and designed the experiments, performed the experiments, analyzed the data, contributed reagents/materials/analysis tools, prepared figures and/or tables, authored or reviewed drafts of the paper, approved the final draft.
- Olga Reñones conceived and designed the experiments, performed the experiments, analyzed the data, authored or reviewed drafts of the paper, approved the final draft.
- Adam Gouraguine performed the experiments, analyzed the data, authored or reviewed drafts of the paper, approved the final draft.
- Andrew F. Johnson performed the experiments, authored or reviewed drafts of the paper, approved the final draft.
- Joan Moranta conceived and designed the experiments, performed the experiments, analyzed the data, contributed reagents/materials/analysis tools, authored or reviewed drafts of the paper, approved the final draft.

## Data Availability

  All raw data is available in Supplemental Files.

## Supplemental Information

Supplemental information for this article can be found online at http://dx.doi.org/10.7717/peerj.6797#supplemental-information.

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
