# Peer review of "Fish nursery value of algae habitats in temperate coastal reefs"

_PeerJ, doi:10.7717/peerj.6797_

## Round 0.1 · original submission · Minor Revisions

Dear Authors. The study "Fish nursery value of different algae habitats within temperate coastal reefs" is interesting and well performed and fills a gap of knowledge in the field of temperate marine ecology. This study contributes with important findings but this manuscript should be revised for clarity and several specific minor revisions, as requested below, should be done.

·

Basic reporting

The manuscript entitled " Fish nursery value of different algae habitats within temperate coastal reefs " by Hinz et al., analyses the main environmental and biological factors in shallow rocky bottoms that determines the success of recruitment of some key species of fish. The topic is interesting and the paper is a good contribution to an important aspect of the ecology of fish assemblages, essential for its adequate management and protection. The experimental design is adequate, the ms is well structured and organized and the results and conclusions are pertinent and relevant.
Therefore, I consider that the work is worthy to be published.

There are however some minor considerations that must be taken into account.

In the material and methods section, the explanation for the interpretation of PCA graphs is perhaps too detailed for a quite generalized analyses, it could be summarized and referenced.

By the contrary, the sampling design is a little bit ambiguous and could be better explained. While the general idea of the design is clear, however the sentence in line 154-156 “Some of the sampling stations were sampled in both spring and winter although focusing on settlers of different species.”, suggest that not all the samples have been used for the same analyses. This aspect should be better explained.

There are some typing mistakes, for example “Cystoceira” at line 125

There are some missing references, for example in lines 230, 246…

Some expressions should be reconsidered, for example, in line 308 “Cystoseira spp. (1-ET) decreased scientifically in height between spring and summer surveys”. It is clear that science has nothing to do with the growth of Cystoseira. Authors must better explain that the results are statistically significant.

At the end of the discussion, some considerations are confusing. It is clear that the suitability of a habitat for the maintenance of a species should not only consider the settlement of juveniles or the abundance of adults of any age, and should also consider its suitability for recruitment, but one thing should not exclude the other.

The work can have an impact since it is clear and concise and provides relevant information to understand the recruitment processes in fish populations in shallow rocky habitats

Experimental design

no comment

Validity of the findings

no comment

Reviewer 2 ·

Basic reporting

This article is an interesting approach to the role of habitat to three benthic fish settlement and juvenile survival.
The introduction is well written and focused correctly in the main issues of the study. Nevertheless, there are some aspects that should be clarified with a more detailed explanation or including some references of previous studies for a better comprehension.
Line 71: In the sentence “There survival of juvenile settlers from recruitment trough the adult life stages..” there is a little confusion between settlement and recruitment, since as it is seems that the juvenile settlers come from recruitment.
In the line 385 the authors use the term juvenile settlement, which is confusing since “juveniles” are considered already recruits, and thus have overcome the settlement and post-settlement mortality.
The article is well written, but with some typographical errors which should be corrected:
Line. 22- This instead of these
Line 89, 221, 352, 524- space needed
Line 125- dobble point at last of the sentence
Line 241- (reference) need to insert a reference for this sentence
Line 319- “scientifically” does not make sense here

The literature references are in general good and well cited. Nevertheless, there are some sentences that should be reinforced or make more clear
Line 61- I would cite also Thierret et al., 2016
Line 68- I would see a reference of some study stating this point
Line 103- Idem
Line 136- In the method section, I would see more information abou the timing period of the studies species, since there are several studies describing diferent periods of settlement, showing a certain variability in the settlement period. This is a major issue since the design of sampling for this study is based on the settlement period of the studies species.
For Diplodus vulgaris the authors do not specify the settlement period of this specie. The autors sample in spring in aims to describe the Distribution of settled individuals on the different habitat types, and also in the discussion (line 399) the authors describe a settlement period for this species during spring (but not specifying which month). But there are several studies describing the settlement period for D. vulgaris in winter, with 2 pulses, the first in November-December and the second in January-February (Vigliola et al, 1998; GarciaRubies & Macpherson 1995, Harmelin-Vivien et al.1995), also an study that the authors cite describe a settlement period for this specie from winter until march (Biaggi et al., 1998).
The same happens for Symphodus ocellatus while some authors (Raventos et al., 2009; 2005) described the settlement period in May and mid-July, not in late summer as other studies do. Other studies that the authors cite for S. ocellatus and Coris julis (Harmelin et al., 1995) do not study this species.
Thus, I believe that this point should be more discussed and clarified.
All the citations should be carefully checked, since there are some citations that do not appear in the Reference section, e.g.: Atlin et al., 2015 (line 139), Kabasakal, 2001 (line 144), Sinopoli et al., 2001 (line 145).
In the line 241 there is some reference missing.

Refecrences:
García-Rubies, A. and Macpherson, E., 1995. Substrate use and temporal pattern of recruitment in juvenile fishes of the Mediterranean littoral. Marine biology, 124(1), pp.35-42.
Harmelin-Vivien, M.L., Harmelin, J.G. and Leboulleux, V., 1995. Microhabitat requirements for settlement of juvenile sparid fishes on Mediterranean rocky shores. In Space Partition within Aquatic Ecosystems (pp. 309-320). Springer, Dordrecht.
Raventós, N. and Macpherson, E., 2005. Effect of pelagic larval growth and size-at-hatching on post-settlement survivorship in two temperate labrid fish of the genus Symphodus. Marine Ecology Progress Series, 285, pp.205-211.
Raventós, N. and Macpherson, E., 2005. Environmental influences on temporal patterns of settlement in two littoral labrid fishes in the Mediterranean Sea. Estuarine, Coastal and Shelf Science, 63(4), pp.479-487.
Vigliola, L., Harmelin-Vivien, M.L., Biagi, F., Galzin, R., Garcia-Rubies, A., Harmelin, J.G., Jouvenel, J.Y., Le Direach-Boursier, L., Macpherson, E. and Tunesi, L., 1998. Spatial and temporal patterns of settlement among sparid fishes of the genus Diplodus in the northwestern Mediterranean. Marine Ecology Progress Series, 168, pp.45-56.

Experimental design

This study contains a high amount of work, which is well designed and described.
I have only one aspect that maybe the authors can clarify. This is the algal sampling and the relationship between different algae morphotypes and juvenile fish abundances. It is not clear for me how they correlated the algal cover of the three 0.25 m2 quadrats with the 30m2 fish transects. Did the authors considered that the algal coves is homogeneous in all the transect? If not, did they observed some variability in a small scale inside the transect? This may explain the differences with other studies such as Cheminée et al 2013, 2017 and Thierret et al., 2016, which were performed in the same area of study. Maybe a deeper discussion on this aspect could clarify this point.

Validity of the findings

The findings of this study are interesting, especially the diet study of fish, that brings new knowledge for these species. For the relationship with juvenile fish and algae, in which did not find a clear pattern, the authors could better clarify comparing with the results of other similar studies performed in the area.

Reviewer 3 ·

Basic reporting

Some spelling and editing mistakes (also in attachments and references), but no big worries. The references are OK perhaps you can have a look on the studies on tropical seaweed beds as fish nurseries by Tano et al. and also some studies from the great barrier reef. Structure is OK, a bit wordy in the introduction and in the results. The hypothesis could be more elaborated and stated more clearly. The study has nothing to add when it comes to different aspects when it comes to algae species composition and different algae habitats their potential exchangeability in a changing world as this has not been analysed/presented. However, the ecological functions of a algae habitat is studied and can be discussed. So the authors could make the outcome of the study stronger by elaborate such message.

Experimental design

The research question could be tighten and more clearly stated (see above). The research is relevant and meaningful and do fill an knowledge gap, and the investigation seems of high standard. However, some clarifications when it comes to choice of studied species and description of sites should be clarified for increased quality (see specific comments). The lack of specific algae species information and other juveniles then specific studied fish species are to be discussed and explained, as well as the choice of study depths. Also seasonal and diurnal variations should be managed.

Validity of the findings

In line with comments above the findings are relevant and interesting however, there is a lack of further discussion putting the results into a wider perspective. What will happen with this fish species if the Cystoseira dominated habitats (or "Cystoseira-like" habitats ) will continue to decrease?

Additional comments

The study is interesting and well performed and do fill a gap of importance in the temperate marine areas. The importance of vegetated rocky shores are stated but not understood as being under studied. The study do contribute with important findings however, the paper can be and should be revised for clarity (aim, methods and discussion) for its fully potential.

Annotated reviews are not available for download in order to protect the identity of reviewers who chose to remain anonymous.

---

## Round 0.2 · accepted · Accept

This is a very interesting research. Thank you for your submission.

# Reviewer 2 ·

Basic reporting

The corrections made by the authors clarify my doubts about the contents of the article and correct typographical and grammatical errors.

Experimental design

The corrections made by the authors clarify my doubts about the contents of the article and correct typographical and grammatical errors.

Validity of the findings

The corrections made by the authors clarify my doubts about the contents of the article and correct typographical and grammatical errors.

Additional comments

The corrections made by the authors clarify my doubts about the contents of the article and correct typographical and grammatical errors.